# The Effects of Temperature and Water on the Seed Germination and Seedling Development of Rapeseed (*Brassica napus* L.)

**DOI:** 10.3390/plants11212819

**Published:** 2022-10-23

**Authors:** Asma Haj Sghaier, Ákos Tarnawa, Hussein Khaeim, Gergő Péter Kovács, Csaba Gyuricza, Zoltán Kende

**Affiliations:** Institute of Agronomy, Hungarian University of Agriculture and Life Sciences, 2100 Gödöllő, Hungary

**Keywords:** abiotic stresses, germination test, oilseed rape, priming, seed number, temperature, water

## Abstract

The seed germination and seedling growth of rapeseed are crucial stages in plant life, especially when facing abiotic stresses. In the present work, the effects of water and temperature on seed germination and seedling growth were investigated in a rapeseed crop (*Brassica napus* L.). The plants were examined under different temperature levels (5 °C, 10 °C, 15 °C, 20 °C, 25 °C, 30 °C, and 35 °C) and water levels (twenty-nine levels based on either one-milliliter intervals or as a percentage of the thousand-kernel weight (TKW)). Moreover, planting densities and antifungal application techniques were investigated in the study. The findings demonstrated substantial variations between all the growth parameters investigated at all the tested temperatures, and 20 °C was considered the optimum within a broad range of 15–25 °C. Water availability plays a significant role in germination, which can be initiated at 0.65 mL, corresponding to 500% of the TKW. The method of TKW is a more accurate aspect of water application because of the consideration of the seed weight and size. The optimal water range for the accumulation of dry weight, 3.85–5.9 mL (2900–4400% of TKW), was greater than that required for seedling growth, 1.45–3.05 mL (1100–2300% of TKW). Twenty to twenty-five seeds per 9 cm Petri dish exhibited the most outstanding values compared to the others, which provides an advantage in breeding programs, especially when there are seed limitations. Seed priming is a more effective antifungal application strategy. These data can be incorporated into future rapeseed germination in vitro studies, breeding programs, and sowing date predictions.

## 1. Introduction

Rapeseed (*Brassica napus* L.), generally known as oilseed rape, is one of the world’s most essential and prolific oilseed crops [1]. In 2020, rapeseed was cultivated on 35.49 million hectares globally, yielding 72.37 million tons of seeds [2]. It is the second most crucial oil crop [3,4]. Rapeseed is used on a small scale for animal feed and lubricants and in paint industries and the bioenergy industry, and it is primarily grown for its edible oil. Due to its high oleic acid (approximately 60%) and linolenic acid (omega 3, ca. 10%) contents, rapeseed oil provides more health advantages to humans than any other oilseed crop [4,5]. Furthermore, because of its high protein content, oilseed rape meal is used as animal feed [6].

Seed germination is the initial stage of a plant’s life. It is a three-phase process. The first phase is imbibition. Dormant seeds take up water, and the hydrolysis process takes place. Seed imbibition results from the interaction of proteins, carbohydrates, and lipids, and variations in their contents can affect the process. Protein and oil bodies are the primary reserves in oilseed crops that provide energy, carbon, and nitrogen to seedlings during their establishment [7]. The second phase is the regulation of germination, characterized by the activation of ATP synthesis in glycolysis, the Krebs cycle, respiratory chain, and the translation of stored mRNA. However, the third phase represents the completion of germination, when the radicle protrudes from the seed coat and forms a root, and the plumules form a shoot system capable of utilizing inorganic matter, water, and light energy for healthy growth [8,9]. Germination is a complicated process from a physiological standpoint, involving multiple signals, and it is influenced by both intrinsic and extrinsic factors [10]. Intrinsic factors include seed dormancy and available food stores, and extrinsic factors include water, temperature, oxygen, light, and relative humidity [11,12,13].

Water is considered the primary germination regulator, as germination begins with seed imbibition. Sufficient moisture must be present for germination to take place. Some research studies stated that a lack of water availability is the primary limitation affecting seed germination [14,15]. It is necessary for enzymatic reactions and the mobilization of the seed storage reserves, including lipids, carbohydrates, and proteins [9]. Therefore, the depletion of the soil water content influences seed imbibition. Slow water absorption by a germinating seed might threaten its emergence and the subsequent crop stand [16]. Therefore, a water shortage inhibits the enzymes responsible for hydrolyzing endosperm starch, which supplies energy for plant development by metabolizing sugar [17]. Water stress, in general, is followed by oxidative stress in seeds during germination, which is characterized by an increase in the production of reactive oxygen species [18,19,20,21]. Therefore, a combination of the antioxidant activity defense system and the internal content of these substances are essential for successful germination under limited water conditions [22,23,24]. 

Temperature is a critical environmental factor in seed germination [25,26]. The pace and rate of germination, which govern water absorption, may be affected by temperatures above or below the optimal range. Under optimal conditions, the absorption process is fast. Some research studies showed that the number of germinated seeds increases linearly as the temperature rises to an optimal level and then decreases linearly as the temperature exceeds the limit [27,28]. In addition, the temperature has a substantial impact on both biochemical and physiological metabolic processes. The latter can regulate enzymatic activity and biochemical reactions during the germination initiation process. Low temperatures reduce the activity of enzymes and slow down food mobilization, limiting the metabolic processes necessary for germination and development [9]. 

A good performance of seed germination results in increased plant density, which is critical for the rapeseed yield and stability in direct sowing plantations [29]. Generally, increasing the plant density increases seed production. The recommended plant density for hybrid rapeseed is approximately 60–70 plants m^−2^ in Europe. However, the average seed production can be obtained within a wide range of plant densities, varying from 8 to 90 plants m^−2^ [29]. The establishment of a high-quality crop and the subsequent crop stand performance are primary objectives of farmers in ensuring a high crop productivity and profit. In addition, the abiotic stressors, including drought, extreme heat, and salinity, can all have detrimental effects on plant growth, resulting in yield loss and a reduced crop quality [30], which can have severe economic consequences for farmers. Water and temperature stresses are the primary constraints on successful crop establishment and the subsequent crop performance. Therefore, unfavorable circumstances might severely affect the establishment of the crop and subsequent yield [31,32,33].

The southern and central parts of Europe have become more prone to drought and temperature extremes over the last two decades. Resistant crops cannot withstand prolonged periods of abiotic stress. Although winter rapeseed crops are typically sown between mid-August and early September, their emergence may be delayed and the yield may be reduced if precipitation does not occur within 10 to 14 days of sowing [34,35]. Extreme temperatures, combined with water scarcity, may contribute to the development of high-temperature stress. In addition, rapeseed has tiny seeds and is susceptible to desiccation when subjected to severe abiotic stress. As a cool season winter crop, oilseed rape is temperature- and water-sensitive [34,35]. 

In scientific studies aimed at enhancing seed germination and crop establishment, pre-sowing seed treatments and agronomic practices have garnered increased attention [36,37,38]. To our knowledge, the effects of water, temperature factors, the seedling density, and pre-treatments on seeds are limited in the case of in vitro rapeseed germination. Due to the importance of rapeseed cultivation for the social and economic life of Hungary, as well as the importance of this crop as an oilseed crop with significant nutritional value, it is essential to determine the optimal conditions and agronomic practices for growing rapeseed in vitro and under different climatic conditions. In this context, the objectives of this study were: (i) to assess the effect of temperature on rapeseed germination and seedling growth and determine the optimal temperature for germination; and (ii) to determine the amount of water required for seed germination using a volume of water in one-milliliter intervals and the thousand-kernel weight (TKW). This technique was recently used to optimize the amount of water required for seed germination in wheat and maize crops [39,40]. It was proved that the water requirements of the seeds differed depending on their size and weight. In the literature, seed size variation may result in variability in the seed water relations and the ability to emerge from a variety of sowing depths [41], as well as resistance to moisture stress [42,43,44]. Our last aim was (iii) to determine the effects of the seed number and density of the seedlings, as well as the effect of pre-sowing seed technology, used to control fungal growth, on seed germination and seedling growth. Germination tests using varying water and temperature levels and seed numbers can establish all the conditions for successful rapeseed germination regardless of the environmental factors, sowing date, or plant density in vitro. Therefore, understanding the optimal temperature and amount of water required for oilseed rape seed germination could aid in the development of an efficient germination program and production technologies.

## 2. Results

### 2.1. Temperature Trial 

The germination of the oilseed rape was conducted at temperatures ranging from 5 °C to 35 °C across the germination time course, with 5 degrees Celsius intervals (Figure 1). Germination was detected approximately three days after the experiment began, and successful germination occurred on average after four days at temperatures of 15 °C, 20 °C, 25 °C, and 30 °C. The oilseed rapeseed appeared to germinate within a range of temperatures, with 20 °C being the most suitable and optimal temperature for obtaining a high germination percentage. At 5 °C and 10 °C, the oilseed rape seeds germinated slowly, taking approximately 9 and 13 days, respectively, and germination was poor at 35 °C.

The recorded data of the radicle and shoot growth under different temperature levels were measured when 80% of the seedlings reached 1 cm in length. Radicles and shoots can grow at a variety of temperatures but are primarily raised at 10 °C, 15 °C, 20 °C, and 25 °C (Figure 2 and Figure 3). The radicle grew optimally at a temperature of 20 °C, while the shoot grew optimally at around 10–20 °C (Figure 2). Given that the radicle grows in soil, it requires a higher temperature than the shoot. As a result, the optimal temperature range for their growth is between 15 and 20 °C. Although the radicle could grow at a suboptimal temperature of 15 °C, it was significantly smaller than it was when it was maintained at 20 °C. Beyond the optimal range, the growth of the two organs accelerated until they reached their maximum height, at which point they stabilized. The shoot and radicle do not require prolonged exposure to elevated temperatures (Figure 2 and Figure 3). In fact, their growth rate was slow for the first four days at 30 °C, and then decreased sharply. Additionally, minimal growth was recorded under 5 °C and an absence of growth was noted at 35 °C. 

The biological effects of various temperatures, ranging from 5 °C to 35 °C, on the growth of the seedlings were recorded daily during the experimental period (Figure 4). The results indicated that the highest rate of growth occurred at 20 °C, followed by 15 °C, 10 °C, and 25 °C (Figure 4). A temperature of 20 °C was suitable for seedling growth. Similar growth patterns were observed at 10 and 25 °C, but the seeds germinated earlier under 25 °C than at 10 °C (Figure 1). At 5 °C, the seedling growth was stable but slightly inhibited, necessitating additional time for development. Additional temperature elevation beyond the optimal range was found to be detrimental to seedling growth, as evidenced by the decreased length value at 35 °C. Indeed, the seedlings grew more rapidly during the first few days at 30 °C, but their growth pattern deteriorated after a few days when they were exposed to prolonged periods of high temperatures (Figure 4). 

### 2.2. Water Trial

The summarized results for the growth indices and dry weight accumulation in response to two water levels, based on a single milliliter and the TKW (thousand-kernel weight) percentage, revealed significant differences in the length of the seedlings’ radicles and shoots between different water levels (Figure 5 and Figure 6; Table 1 and Table 2).

Oilseed rape can be germinated at low water levels of 0.65 mL or 500% of the TKW (Table 2). The optimal range for the germination of twenty seeds per Petri dish was approximately 1.45–3.45 mL, using 500 and 4400% of TKW (Table 2). 

The radicle growth increased significantly as the water quantity increased to the optimal level but decreased significantly as the water quantity increased beyond the optimal level (Figure 1 and Figure 2). The optimal water range for radicle growth was 1.45–1.85 mL based on the TKW, which was within the optimal range determined using the milliliter-based method (Table 1 and Table 2). The plumule exhibited a similar growth pattern to the radicle but with a more extensive optimal growth range of 3.45–5.1 mL, representing 2600–3800% of the TKW. 

As a result, the measurement of the water requirement for the whole seedling is more accurate. Thus, using the TKW percent method, the optimal range for seedling growth was approximately 1.45–3.05 mL, representing 1100–2300 percent of the TKW (Table 2). In addition, it fell within the optimal range established by the one-milliliter (1–4 mL) water-based method (Table 1). Therefore, it can be stated that the TKW method is more precise in optimizing the water requirements for germination.

The dry weight, or dry matter, is a critical parameter that is primarily used to describe water use efficiency, estimate yields, and select drought-tolerant plants. An analysis of variance was performed based on the dry weights of the shoot, radicle, and total seedling, as well as the corrected dry weight of the total seedling, which was calculated by subtracting the number of non-germinated seeds (Table 1 and Table 2). The results showed significant differences in all the tested parameters across the various water levels. The dry weight of all the radicles and shoots increased significantly as the water volume increased, peaked at the optimal level, and then decreased slightly despite the increased water volume (Figure 5 and Figure 6; Table 1 and Table 2). The optimal water range for the accumulation of dry weight by the seedlings was evaluated to be 3.85 to 5.9 mL, corresponding to 2900 to 4400% of the TKW, which was greater than the optimal range required for seedling growth (1.45–3.05 mL or 1100–2300% of the TKW). The shoot (3.85–5.5 mL) accumulated more dry weight than the radicle (1.05–1.85 mL), given the radicle’s delicate structure and lower weight. The radicle length and dry weight were reduced at levels of more than 6 mL (Figure 5 and Figure 6), indicating that the plant was susceptible to waterlogging.

### 2.3. Seed Number Trial

The results indicated a significant difference only in the subdivision of seedlings with healthy shoot growth among which the aggregated values of 5, 20, and 25 seeds per Petri dish were used for the seed number test (Table 3). However, this significantly affected the final aggregated value. The seedlings with short plumule growth and with radicle growth alone, non-germinated seeds, and seeds that were initially germinated did not vary significantly as the seed number increased (Table 3). However, a higher aggregated value was observed when using 25 seeds per Petri dish. Therefore, densities of 20–25 seeds moistened with 5 mL of water appeared to be the optimal density for growing an oilseed rape crop in vitro. 

### 2.4. Antifungal Trial 

The recorded data and comparative findings for germinating seeds pre-treated in two growth solutions containing antifungal Amistar Xtra and Hypo solution (10% Sodium hypochlorite (NaClO)) showed a significant effect on the growth parameters compared to the control (Figure 7). The germination of seeds primed with Hypo significantly affected the radicle and seedling growth compared to the control, whereas the antifungal Amistar Xtra growth media inhibited fungal growth as well as the radicle, shoot, and seedling growth. The antifungal Amistar Xtra used in the growth media had a negative impact on the radicle, shoot, and seedling growth (Figure 8). In fact, all the growth parameters measured here decreased proportionately as the antifungal concentration increased, even at a low concentration.

The techniques of seed priming with fungi sterilizer solution and growing of seeds in growth media containing the fungicide Amistar Xtra were compared in this experiment (Figure 9). The priming technique showed a significantly more positive effect on all the growth parameters than the amendment of seeds in the growth media. Seed pre-treatment or priming can offer a viable alternative for the prevention of fungal growth during seed germination in vitro.

## 3. Discussion

Temperature is critical for regulating plant growth and development [26]. The rate of the rapeseed germination potential increases linearly as the temperature rises and then decreases linearly until reaching the optimal level, followed by a reduction beyond that level [28,45]. In the current study, oilseed rape germinated at temperatures ranging from 5 °C to 30 °C, and 20 °C was the optimal temperature for germination within a broad range of 15–25 °C (Figure 3). This finding is consistent with another study [46]. A temperature above or below the optimum caused the germination potential to drop. High temperatures of 35 °C greatly hindered seed germination and subsequent emergence owing to the inhibitory impact of high temperatures. However, low temperatures of 5 °C or below caused delayed germination, so that the process required a longer time (Figure 1). This finding is consistent with previous findings [47,48].

The optimal temperature of 20 degrees Celsius led to the most rapid and complete germination of seeds after four days of incubation (Figure 1). The increased temperature accelerated the germination rate [49,50]. However, germination was sluggish when the temperature was reduced to 5 °C. Metabolic reactions and enzyme activity cause variation in the germination time during the germination process. Indeed, lower temperatures cause seeds to have a slower metabolism, resulting in slower growth, whereas higher temperatures cause plants to have a faster metabolism, dissipating the seed energy required for growth. According to enthalpy approaches, as the temperature rises, the energy in the water increases, resulting in an increase in diffusion pressure, which simultaneously increases metabolic and enzyme activity and decreases the internal potential of a seed, thereby promoting increased water absorption. Thus, hydration occurs more rapidly at higher temperatures, a physical process that may accelerate germination [51]. At a super-optimal temperature, the available energy in the seed’s cellular members remains unfavorable for embryonic growth and quickly dissipates [49].

Soil temperature is a critical environmental factor affecting the growth and development of roots. When the soil temperature reaches the optimal level, root growth increases. However, root growth declines when it exceeds the optimal level [52]. At 20 °C, a rapid growth pattern with a higher growth value was observed; therefore, this was considered the optimal temperature for radicle growth. The same pattern was observed at 10 °C and 15 °C, but with lower values (Figure 2). Additional temperature elevation was detrimental to radicle growth, as evidenced by the low average value at 35 °C. At 25 °C, the radicle grew slightly for a few days before ceasing to grow and stabilizing. At 30 °C, the initial growth of the radicles was more significant compared to the subsequent decrease after a few days (Figure 2). The radicle’s sensitivity to high temperatures results from the cumulative temperature requirement for each growth stage. Surpassing the optimal temperature results in quicker germination, but this is not the case throughout the development and growth period (Figure 1). These findings corroborate previous research [39,40]. The radicle requires different cumulative temperatures throughout its growth stage. Not only does germination require a specific temperature, but each stage has its own. As a result of the complexity of the germination process, the temperature response may vary throughout the germination period [53]. In addition, as oilseed rape seeds are susceptible to damage under prolonged high temperatures, prior research has often focused on heat shock and gradual temperature rises from the ideal to a higher value [54,55]. They concluded that the alternated heat accumulation potential stimulated by gradual temperature stress might be more important than heat tolerance induced by constant or unexpected heat exposure [54]. The radicle cannot grow for an extended period of time at constant higher temperatures, a finding that is comparable to that which we observed at 30 °C (Figure 2).

The shoot can grow in a temperature range of 10–25 °C, with a minor difference between different temperature levels (Figure 3). Compared to the radicle, the shoot grew in a different growth pattern (Figure 3). The radicle’s optimal temperature was 20 °C, while the shoot’s optimal temperature was between 15 °C and 20 °C regardless of its growth stage (Figure 3). The radicle requires a higher temperature than the shoot due to its internal organs, which grow into the soil. During the very late stage of the germination process, the steady growth of the shoot was observed at temperatures of 20 °C and 25 °C (Figure 3). The seedling growth was significantly influenced by temperature. The temperature of 20 °C was proven to be the optimum temperature for seedling development (Figure 4). Most enzymes become inactive at temperatures over 35 °C, negatively impacting germination and seedling development [56].

Water intake is a prerequisite for germination. It is required for seed imbibition, enzymatic activation, degradation, translocation, and the utilization of reserve storage material. The germination capacity increased significantly as the water volume increased until it reached the optimal level, and then it decreased slightly as the water level increased (Figure 5 and Figure 6). Under conditions of limited water availability, seed imbibition is insufficient for initiating germination. However, increased water availability results in waterlogging, which inhibits seed germination due to oxygen depletion. The dormant seed requires water, oxygen, and an appropriate temperature to complete its life. According to the TKW method, the seeds started germinating at a water content of 0.65 mL, corresponding to 500% of the TKW (Table 2). This amount of water may be very close to the moisture content required for germination demands.

Most seeds require a critical moisture content in order to germinate. This value was estimated to be 30% in maize, 40% in wheat, and 50% in soybeans [53]. Once the critical seed moisture content is reached, sufficient moisture is present to initiate germination. Moreover, numerous studies have documented the minimum water potential required to induce seed imbibition [15,57]. Seed imbibition is a three-part process, commencing with an initial phase of water intake, followed by a plateau period with a minor change in the water content, and concluding with an increase in the water content, as evidenced by radicle development [58]. Generally, seed germination regulation occurs during the plateau period, as germination is complete when the embryo begins to expand. The duration of this plateau phase is dependent on the water potential. Therefore, the duration of this phase is determined by water availability and the species. It can be extended at water potentials close to −0.03–1.00 MPa in some crops [15,57].

According to the TKW method (Table 4), seeds can be germinated over a range of water levels, starting at the wilting moisture point (0.65 mL), demonstrating the TKW method’s accuracy in detecting and optimizing the water requirement for seed germination, as previous studies have reported [39,40]. Indeed, seeds require a hydric range within which they can either not germinate or germinate poorly or slowly [59]. Waterlogging, which occurs because of an increased water volume, depletes the oxygen available to seeds, which is a critical factor in seed germination [60].

The ideal level of water for germination has a significant impact on the subsequent development of seedlings. The seedling length increased significantly with increasing water availability until it reached its maximum value and decreased significantly with increasing water availability. The shoot and radicle structures exhibited distinct optimal water ranges for growth and development. Under non-limiting hydric circumstances, the radicle expanded slowly and encouraged shoot development. As a result, the measurement of the total seedling length appeared to be more accurate in determining the optimal water demand for the entire plant, which was determined to be 1.45–3.05 mL or 1100–2300% of the TKW (Table 2). Under ideal circumstances, seedlings developed rapidly, which is consistent with previous research [57]. Radicle growth is the most critical factor in early seedling survival because it enables the seedling to exploit water deep in the soil via rapid root extension [61].

The dry weight accumulation at various water levels revealed a significant difference in the dry weight of the radicles and shoots (Table 1 and Table 2). The optimal water range for the seedling’s dry matter was 3.85–5.9 mL, corresponding to 2900–4400 % of the TKW. Increased water availability had a detrimental effect on dry matter accumulation. The radicle is more sensitive to high water availability, requiring less water to build up a unit of dry matter. Under optimal conditions, rapid radicle growth contributed significantly to the shoots’ dry matter accumulation. According to functional balance theory, the plant will respond to increased water availability by increasing the flow and assimilating the shoot in order to increase its dry matter [62]. In conclusion, given the effect of the seed size on the internal seed moisture requirements, the TKW water application method enables the determination of the minimum and optimal water requirements for germination and seedling establishment. The current findings are supported by previous research [39,40].

A significant difference between seedlings with healthy shoot growth subjected to the seed number treatment of 15, 20, and 25 seeds per Petri dish was found (Table 3). As the number of seeds per Petri dish increased, the subdivision of seedlings with healthy shoot growth and the aggregated values increased. Under controlled conditions, a seed density of 20–25 seeds per Petri dish is optimal for growing oilseed rape in vitro. Although the density of 15 seeds stimulated germination, the seedlings grew slower. At a low seed number, the lack of seedlings could be attributed to the scarcity of a critical resource, such as water, due to competition [63]. Additionally, dense seedlings are frequently more susceptible to lodging, which increases the rate of disease incidence and, as a result, the seedling emerging percentage [64]. Therefore, optimizing the seed number per Petri dish in germination is a critical consideration during germination, along with other environmental factors.

Seed treatment with fungicides is widely used to improve the seed emergence and resistance to seed- and soilborne fungal pathogens [38], and it has received considerable attention in several recent publications [65,66]. Given that the antifungal Amistar Xtra’s primary function is to inhibit fungal growth, it had a detrimental effect on the radicle, plumule, and seedling growth (Figure 7). Additionally, as the antifungal concentration in the growth media increased, the inhibitory effect became more pronounced (Figure 8). This is possibly because no fungi were present in our experiment rather than because of the fungicide’s phytotoxicity, as many authors explained in their works [65,67,68]. They observed a recovery of seedling growth following fungi exposure and inoculation. Amistar Xtra fungicide, which contains the active ingredient azoxystrobin, inhibits electron transport in the mitochondrial respiratory chain of fungi, thereby decreasing aerobic energy production and inhibiting fungus growth [69]. As a control, seed treatment with Hypo solution had a beneficial effect on the seedling growth. Priming seeds with Amistar Xtra or Hypo rather than the amendment of seeds in antifungal growth media was found to inhibit fungal growth in vitro (Figure 7 and Figure 9).

## 4. Materials and Methods

This study examined the effects of abiotic stressors (water and temperature), seedling density, and fungal growth control on seed germination and seedling growth in vitro. The experiment was undertaken in 2022 at the Hungarian University of Agriculture and Life Science/Institute of Agronomy. Rapeseed (*Brassica napus L. Allison*) was chosen for this study. “*Allison*” is a highly productive hybrid with a high oil and glucosinolate content. In 2015, this hybrid was registered in France. It is a hybrid resistant to *Turnip Yellowing Virus (TUYV)* and has a tall stem, although it is susceptible to lodging and *Cylindrosporium*.

### 4.1. Temperature Stress Trail

The germination ability was tested at seven different temperature levels (5 °C, 10 °C, 15 °C, 20 °C, 25 °C, 30 °C, and 35 °C). Twenty seeds of rapeseed were placed in a standard 9 cm Petri dish with a single filter paper moistened with 5 mL of distilled water in four replications. The Petri dish was sealed with parafilm and subjected to the tested constant temperature in the incubation chambers. The measurement was taken when around 80% of the seedlings in the Petri dishes reached a length of 1 cm (Figure 10). Daily, four Petri dishes were taken out of the chambers at each temperature level for the physical measurement of the length of the radicles and shoots. Germination percentages were also recorded.

### 4.2. Water Stress Trail

The germination ability of oilseed rape under water stress was examined at 29 different water levels. In total, 10 different amounts of distilled water were used based on a milliliter interval of 0–10, and 19 different amounts of distilled water were used on the basis of the TKW (Table 4). At each water level, 100 seeds were placed in five Petri dishes (20 seeds per plate) using filter paper saturated with various amounts of distilled water (Table 1). The following equation (1) was used to compute the amount of water based on the TKW:TKW * Seed n/100,000 = 1% of the proposed water amount(1)

The TKW of the oilseed rape was 6.68 g. Additional information on the calculation of the water quantity based on the TKW is available in two previous studies [39,40]. The Petri dishes were sealed with parafilm to prevent water evaporation and incubated at 20 °C for ten days in a growth chamber. After ten days of incubation, the radicle and shoot lengths and the number of non-germinated seeds were measured. The radicles and shoots were then separated and dried at 65 °C until they reached a constant weight (48 h).

### 4.3. Seed Number Trial

This section of the experiment examined the effects of the seed number on germination and seedling growth using the same volume of water (5 mL) and incubation at a temperature of 20 °C. For the amplification, 15, 20, and 25 seeds were planted in the Petri dishes. Morphological measurements were conducted ten days after incubation, and the measured parameters were divided into five categories. The aggregated germination value (AGG) was calculated using the measured parameters, as described below (Equation (2)) [39,40]:(2)$$\mathrm{AGG}\left(x\right)=\frac{\left(\left(NO-G\times 0\right)+\left(S\times 0.1\right)+\left(R\times 0.25\right)+\left(SP\times 0.65\right)+\left(NS\times 1\right)\right)}{N}$$
where NO g is the non-germinated seed number, S is the number of seeds in which germination started, R is the number of germinated seeds with radicles only, SP is the number of seedlings with a short plumule, NS is the number of normal seedlings, and N is the total number of the tested seeds.

### 4.4. Antifungal Trial

Two different techniques of antifungal application were used to assess the fungicide’s potential to inhibit fungal growth. For the first technique, we added five different concentrations of Amistar Xtra, specifically 0, 20, 200, 2000, and 20,000 ppm, to the growth media. In addition, two different seed sterilization methods were tested, the first based on soaking the seeds for 3 min in a solution of 2000 ppm Amistar Xtra and the second based on a Hypo solution (10% Sodium hypochlorite (NaClO)). After sterilization, the seeds were rinsed with distilled water and incubated for 10 days at 25 °C in a growth chamber. After ten days of incubation, the radicle and shoot lengths were measured, and the germinated seeds were counted.

### 4.5. Statistical Analysis

Analysis of variance (ANOVA) and Fisher’s test of least significant differences (LSD) were conducted for the obtained data. Kolmogorov–Smirnov–Wilk and Shapiro–Wilk tests conducted in SPSS V27 IBM (New York, NY, USA) were used for the data normality verifications. The effects of the water level, seed number, and antifungal treatment on the germination percentage, radicle length, shoot length, and seedling growth were analyzed using a computing program (GenStat twelfth edition, GenSat procedure library release PL20.1m, and MS Excel 365). A sigmoid curve model was applied using a statistical program (JMP Pro 13,2,1 of SAS Institute, Cary, NC, USA, and MS Excel 365) to fit the data and plot the appropriate temperature levels.

## 5. Conclusions

The findings of this study highlighted the critical factors affecting oilseed rape germination and established the optimal range for successful germination and seedling growth. The optimal temperature for oilseed rape germination and seedling growth was 20 °C within a more comprehensive range from 10 °C to 25 °C. Between the optimal and suboptimal ranges, the germination potential decreased. This information could be used to determine planting times depending on the optimal temperature for seed germination. According to the TKW method, oilseed rape seeds can be germinated starting at a volume of 0.65 mL, representing 500% of the TKW and being close to the critical moisture content required for germination. The optimal water range for the accumulation of dry weight, 3.85–5.9 mL (2900–4400% of the TKW), was greater than that required for seedling growth, 1.45–3.05 mL (1100–2300% of the TKW). Therefore, optimizing the water availability for seeds based on the TKW percentage is a more precise strategy, since it considers the seed’s weight and size when estimating the water required to develop optimally.

Additionally, optimizing the seedling density is critical for avoiding water factor limitation and competition between seedlings and minimizing disease problems. Thus, it was determined that a seed density of 20–25 seeds per Petri dish was optimal for growing oilseed rape in vitro. Finally, seed priming with an anti-fungicide to inhibit fungal growth is recommended for the purpose of antifungal control. These findings have the potential to be beneficial in future research and breeding programs.

## Figures and Tables

**Figure 1 plants-11-02819-f001:**
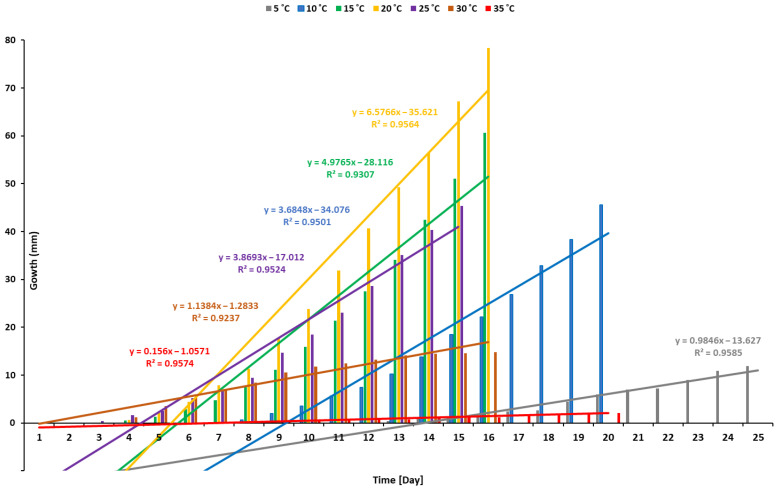
Recorded data on the germination time of rapeseed crop at different levels of temperature.

**Figure 2 plants-11-02819-f002:**
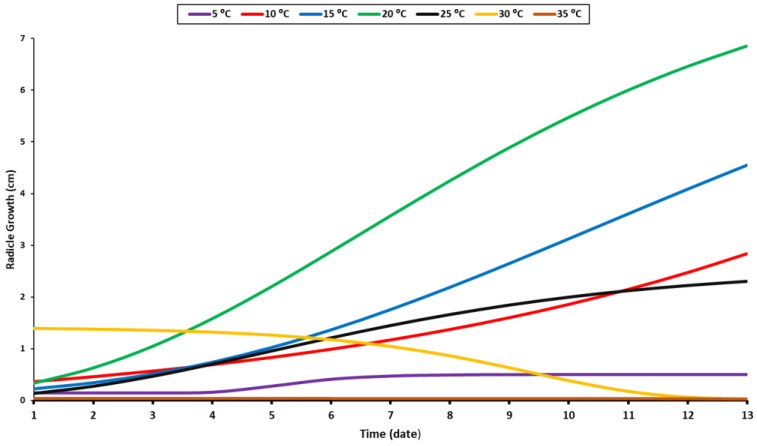
Growth response of radicle to temperature levels.

**Figure 3 plants-11-02819-f003:**
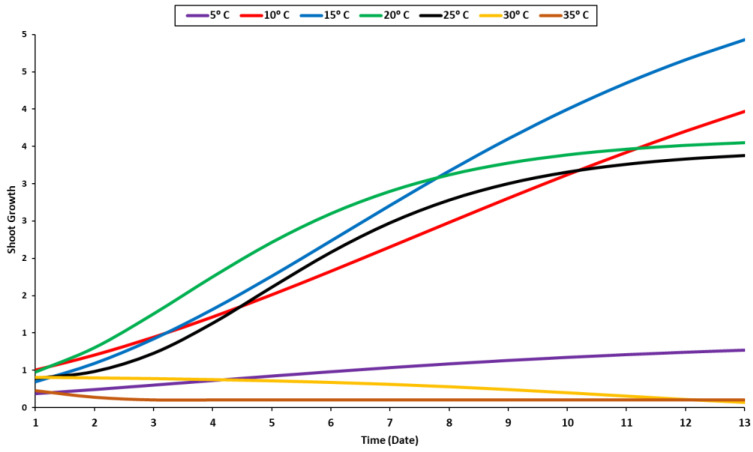
Growth response of shoot to temperature levels.

**Figure 4 plants-11-02819-f004:**
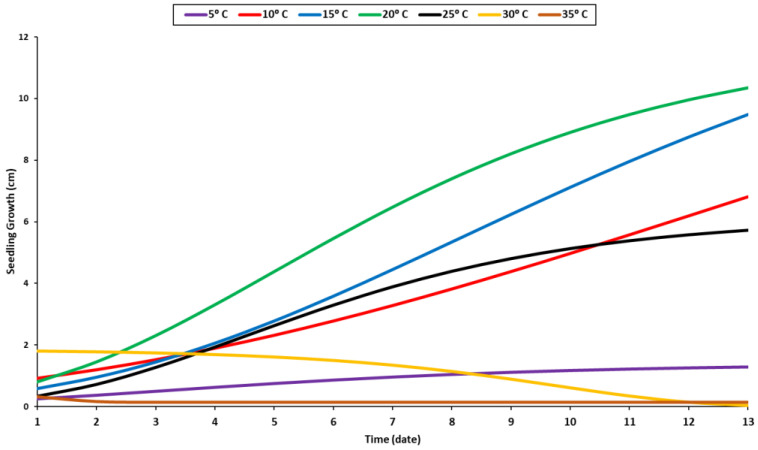
Growth response of seedlings to temperature levels.

**Figure 5 plants-11-02819-f005:**
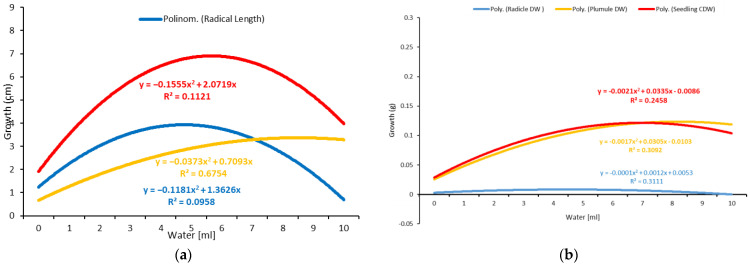
Response of seedlings, plumules, and radicles to different amounts of water tested. (**a**) Growth vs. different amounts of water based on one-milliliter intervals. (**b**) Dry weight accumulation vs. different amounts of water based on one ml intervals.

**Figure 6 plants-11-02819-f006:**
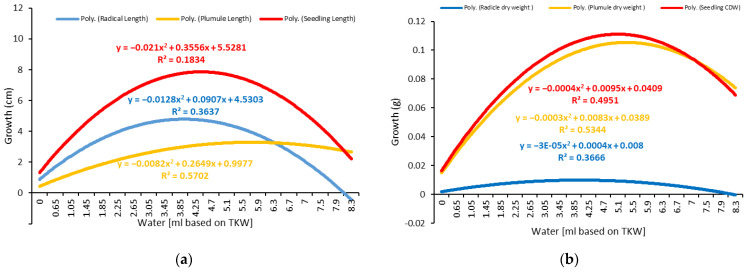
Response of seedlings, plumules, and radicles to different amounts of water at 20 °C. (**a**) Growth vs. water based on TKW%. (**b**) Dry weight accumulation vs. water based on TKW%.

**Figure 7 plants-11-02819-f007:**
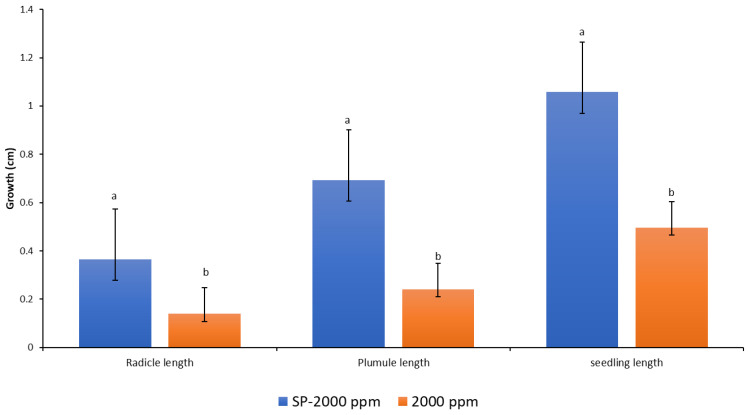
Response of the radicles, plumules, and seedlings to media amended with anti-fungicide (Amistar Xtra) and Hypo solution SP. Values denoted with different letters are significantly different at *p* < 0.05.

**Figure 8 plants-11-02819-f008:**
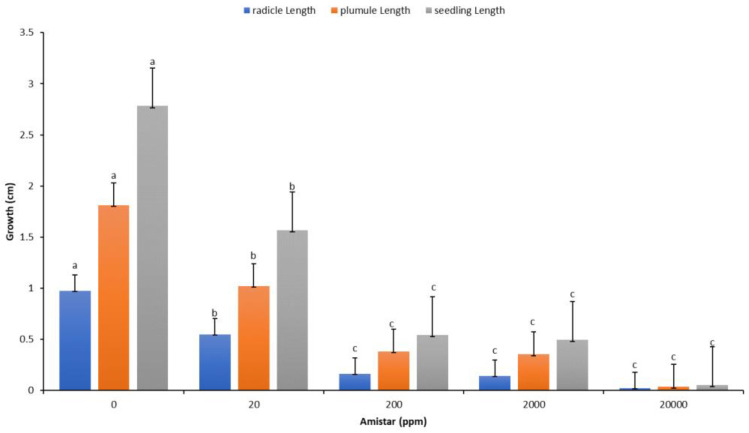
Response of radicles, plumules, and seedlings to different concentrations of antifungal. Values denoted with different letters are significantly different at *p* < 0.05.

**Figure 9 plants-11-02819-f009:**
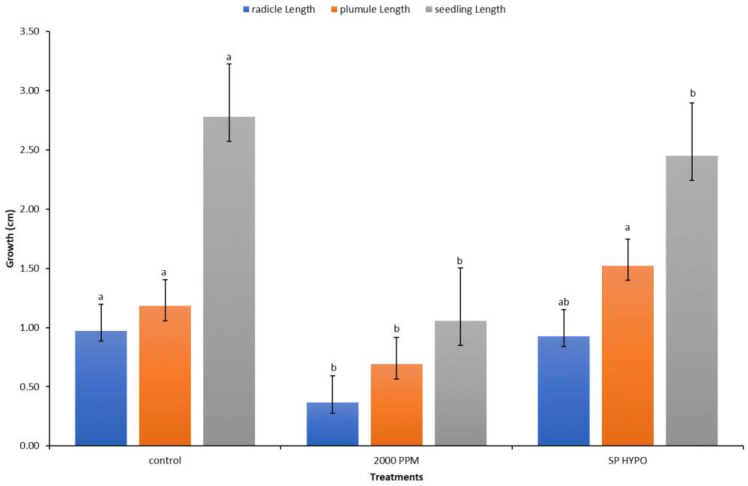
Response of radicles, shoots, and seedlings to two different methods of fungal (sterilization and growth media). Values denoted with different letters are significantly different at *p* < 0.05.

**Figure 10 plants-11-02819-f010:**
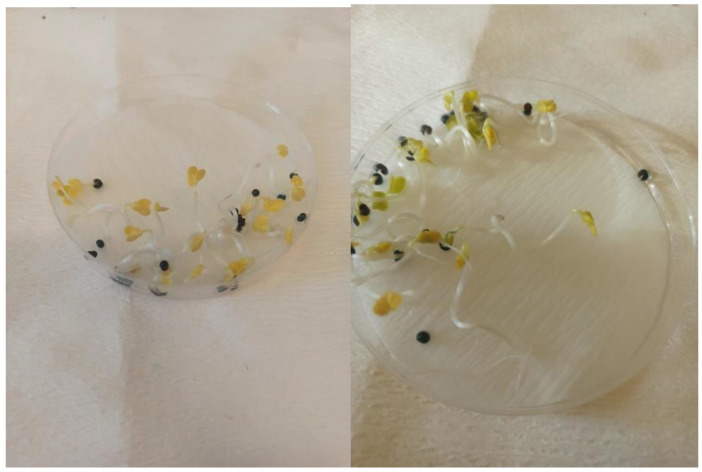
Photos of seed germination in a Petri dish at 20 °C. High values of germination and seedling growth were recorded at this temperature.

**Table 1 plants-11-02819-t001:** Mean data on germination and seedling growth traits for the different amounts of water based on one-milliliter intervals.

Water (mL)	Germinated (n)	Radicle (cm)	Plumule (cm)	Seedling (cm)	Plumule DW (g)	Radicle DW (g)	Seedling DW (g)	Corrected DW (g)
0	0.000 ± 0.00 c	0.000 ± 0.0 d	0.000 ± 0.00 f	0.000 ± 0.00 f	0.000 ± 0.000 f	0.000 ± 0.00 d	0.000 ± 0.000 e	0.000 ± 0.000 d
1	19.800 ± 0.45 a	5.420 ± 1.41 a	1.955 ± 0.29 e	7.375 ± 1.27 ab	0.080 ± 0.005 de	0.012 ± 0.01 a	0.093 ± 0.004 bc	0.092 ± 0.005 b
2	19.400 ± 0.89 a	5.430 ± 0.42 a	2.354 ± 0.14 de	7.784 ± 0.50 a	0.079 ± 0.007 e	0.011 ± 0.01 a	0.089 ± 0.009 bc	0.087 ± 0.012 b
3	19.600 ± 0.55 a	5.888 ± 1.44 a	2.885 ± 0.19 bc	8.773 ± 1.269 a	0.084 ± 0.003 cde	0.011 ± 0.00 a	0.095 ± 0.005 b	0.093 ± 0.006 ab
4	20.000 ± 0.00 a	3.952 ± 1.25 b	3.593 ± 0.38 a	7.545 ± 1.56 ab	0.088 ± 0.011 bcd	0.008 ± 0.00 b	0.096 ± 0.10 b	0.096 ± 0.010 ab
5	19.600 ± 0.55 a	2.780 ± 1.2 bc	2.369 ± 0.49 cd	5.149 ± 1.59 de	0.098 ± 0.002 a	0.008 ± 0.00 b	0.105 ± 0.004 a	0.104 ± 0.003 a
6	16.200 ± 2.28 b	1.648 ± 0.60 c	2.367 ± 0.58 cd	4.015 ± 1.17 e	0.076 ± 0.012 e	0.003 ± 0.00 c	0.079 ± 0.013 d	0.065 ± 0.020 c
7	19.600 ± 0.55 a	1.796 ± 0.45 c	2.925 ± 0.33 bc	4.721 ± 0.73 de	0.092 ± 0.005 abc	0.003 ± 0.00 c	0.095 ± 0.005 b	0.094 ± 0.006 ab
8	19.800 ± 0.45 a	1.718 ± 0.88 c	2.971 ± 0.81 b	4.689 ± 1.65 de	0.094 ± 0.005 ab	0.003 ± 0.00 c	0.096 ± 0.005 b	0.096 ± 0.005 ab
9	19.000 ± 0.71 a	2.306 ± 0.67 c	3.692 ± 0.59 a	5.998 ± 1.18 cd	0.091 ± 0.002 abc	0.004 ± 0.00 c	0.094 ± 0.003 b	0.090 ± 0.006 b
10	17.400 ± 1.67 b	2.482 ± 0.91 c	3.661 ± 0.43 a	6.143 ± 1.29 bcd	0.081 ± 0.008 de	0.003 ± 0.00 c	0.085 ± 0.007 cd	0.074 ± 0.012 c
LSD	1.251	1.202	0.564	1.545	0.008	0.0019	0.0086	0.0119

Different letters indicate a significant difference between treatments at *p* < 0.05, starting sequentially, with the letter (a) being the most significant. Data presented as mean ± SD.

**Table 2 plants-11-02819-t002:** Mean germination and seedling growth values for the different amounts of water based on TKW%.

Water (mL)	Germinated (n)	Radicle (cm)	Plumule (cm)	Seedling (cm)	Plumule ^1^DW (g)	Radicle ^1^DW (g)	Seedling ^1^DW (g)	Corrected ^1^DW (g)
0	0.0 ± 0.00 e	0.000 ± 0.00 j	0.000 ± 0.00 i	0.000 ± 0.00 g	0.000 ± 0.00 i	0.000 ± 0.00 h	0.000 ± 0.00 f	0.000 ± 0.00 g
0.65	19.8 ± 0.45 ab	4.251 ± 1.08 de	1.453 ± 0.22 h	5.704 ± 1.26 de	0.069 ± 0.03 h	0.010 ± 0.00 cd	0.079 ± 0.03 e	0.078 ± 0.03 f
1.05	19.2 ± 0.45 abc	5.960 ± 1.35 bc	2.182 ± 0.44 fg	8.142 ± 1.39 c	0.080 ± 0.00 efgh	0.013 ± 0.00 ab	0.092 ± 0.00 bcd	0.089 ± 0.00 cdef
1.45	20.0 ± 0.00 a	7.202 ± 1.40 ab	2.713 ± 0.13 cdef	9.915 ± 1.46 ab	0.074 ± 0.01 gh	0.012 ± 0.00 bc	0.086 ± 0.01 de	0.086 ± 0.01 cdef
1.85	19.8 ± 0.45 ab	7.754 ± 0.88 a	2.586 ± 0.17 cdefg	10.34 ± 0.94 a	0.078 ± 0.01 fgh	0.014 ± 0.00 a	0.092 ± 0.01 bcd	0.091 ± 0.01 cdef
2.25	20.0 ± 0.00 a	5.310 ± 1.60 cd	2.363 ± 0.17 efg	7.673 ± 1.66 c	0.082 ± 0.01 defgh	0.010 ± 0.00 c	0.092 ± 0.01 bcd	0.092 ± 0.01 cdef
2.65	20.0 ± 0.00 a	5.939 ± 0.84 bc	2.429 ± 0.27 defg	8.368 ± 1.06 bc	0.086 ± 0.00 cdefg	0.012 ± 0.00 bc	0.097 ± 0.00 bcd	0.097 ± 0.00 abc
3.05	20.0 ± 0.00 a	5.900 ± 2.30 c	2.605 ± 0.25 cdefg	8.505 ± 2.49 bc	0.081 ± 0.01 defgh	0.010 ± 0.00 cd	0.091 ± 0.01 bcd	0.091 ± 0.01 cdef
3.45	20.0 ± 0.00 a	3.937 ± 1.17 ef	3.299 ± 0.39 ab	7.236 ± 1.49 cd	0.084 ± 0.01 cdefg	0.007 ± 0.00 f	0.091 ± 0.01 bcd	0.091 ± 0.01 cdef
3.85	19.8 ± 0.45 ab	2.601 ± 0.65 ghi	2.567 ± 0.37 cdefg	5.168 ± 1.00 ef	0.103 ± 0.01 a	0.007 ± 0.00 f	0.109 ± 0.01 a	0.108 ± 0.01 a
4.25	18.6 ± 1.14 bcd	2.310 ± 0.85 ghi	3.106 ± 0.55 abc	5.416 ± 1.37 ef	0.083 ± 0.01 cdefg	0.004 ± 0.00 g	0.087 ± 0.01 cde	0.081 ± 0.01 def
4.7	19.6 ± 0.55 abc	2.909 ± 0.50 fgh	2.882 ± 0.26 bcde	5.791 ± 0.62 def	0.096 ± 0.01 abc	0.008 ± 0.00 ef	0.104 ± 0.00 ab	0.102 ± 0.01 ab
5.1	19.2 ± 0.84 abc	2.219 ± 0.92 ghi	3.336 ± 0.56 ab	5.555 ± 1.43 ef	0.087 ± 0.01 bcdef	0.004 ± 0.00 g	0.091 ± 0.01 bcd	0.088 ± 0.02 cdef
5.5	18.4 ± 3.85 cd	2.053 ± 1.17 hi	2.122 ± 1.13 g	4.175 ± 2.28 f	0.100 ± 0.00 a	0.008 ± 0.00 ef	0.109 ± 0.00 a	0.100 ± 0.02 abc
5.9	19.8 ± 0.45 ab	3.479 ± 0.31 efg	2.444 ± 0.41 defg	5.923 ± 0.63 de	0.099 ± 0.00 ab	0.010 ± 0.00 cd	0.109 ± 0.00 a	0.108 ± 0.00 a
6.3	19.0 ± 1.23 abc	2.836 ± 0.75 fgh	2.976 ± 0.58 bcd	5.812 ± 1.30 def	0.093 ± 0.01 abcd	0.007 ± 0.00 f	0.099 ± 0.01 abc	0.095 ± 0.02 bcde
6.7	18.6 ± 1.14 bcd	1.710 ± 0.58 hi	2.966 ± 0.59 bcd	4.676 ± 1.00 ef	0.090 ± 0.00 abcde	0.004 ± 0.00 g	0.094 ± 0.00 bcd	0.087 ± 0.01 cdef
7	19.8 ± 0.45 ab	1.796 ± 0.45 hi	2.925 ± 0.33 bcde	4.721 ± 0.73 ef	0.092 ± 0.00 abcd	0.003 ± 0.00 g	0.095 ± 0.00 bcd	0.095 ± 0.01 abcd
7.5	19.8 ± 0.45 ab	2.149 ± 0.27 hi	3.617 ± 0.28 a	5.767 ± 0.25 def	0.094 ± 0.01 abcd	0.004 ± 0.00 g	0.098 ± 0.01 bcd	0.097 ± 0.01 abc
7.9	19.4 ± 0.55 abc	1.466 ± 0.76 i	2.890 ± 0.73 bcde	4.356 ± 1.48 ef	0.096 ± 0.01 abc	0.003 ± 0.00 g	0.099 ± 0.01 bcd	0.096 ± 0.01 abcd
8.3	17.8 ± 1.92 d	1.980 ± 0.54 hi	3.430 ± 0.31 ab	5.410 ± 0.83 ef	0.086 ± 0.01 cdefg	0.004 ± 0.00 g	0.090 ± 0.01 cde	0.080 ± 0.02 ef
LSD	1.325	1.271	0.575	1.645	0.013	0.002	0.0129	0.0154

Different letters indicate a significant difference between treatments at *p* < 0.05. They start sequentially, with the letter (a) being the most significant. Data presented as mean ± SD. 1 Dry weight estimated for radicle, plumule, and entire seedling.

**Table 3 plants-11-02819-t003:** Germination ratio and seedling characteristics of rapeseed as a response to the number of seeds per Petri dish.

^1^SN	^2^No g %	^3^S %	^4^R %	^5^SP-Germ %	^6^N-Germ %	^7^Agg-Value
15	0.140 ± 0.12	0.153 ± 0.02	0.007 ± 0.006	0.013 ± 0.009	0.667 ± 0.03 b	0.692 ± 0.034 b
20	0.085 ± 0.07	0.165 ± 0.02	0.005 ± 0.005	0.025 ± 0.013	0.735 ± 0.03 ab	0.769 ± 0.024 ab
25	0.072 ± 0.07	0.104 ± 0.006	0.020 ± 0.009	0.008 ± 0.005	0.792 ± 0.03 a	0.813 ± 0.029 a
LSD	NS	NS	NS	NS	0.091	0.082

Different letters indicate a significant difference between treatments at *p* < 0.05. NS means non-significant difference between the means. Data presented as mean ±SD. ^1^ Number of seeds tested; ^2^ percentage of non-germinated seeds; ^3^ seeds that started germination; ^4^ percentage of seeds that germinated with obvious roots; ^5^ percentage of seedlings with short shoots; ^6^ percentage of seedlings with healthy shoots; ^7^ aggregated value.

**Table 4 plants-11-02819-t004:** Water amount levels based on one-milliliter intervals and TKW%.

Amount of Water Based on 1 mL of Intervals		Amount of Water Based on the TKW			
^1^TN	^2^WA (mL)	^3^TN	^4^PW (%)	^5^WA (mL)	^6^RAW (mL)
1	0	10	500	0.668	0.65
2	1	11	800	1.0688	1.05
3	2	12	1100	1.4696	1.45
4	3	13	1400	1.8704	1.85
5	4	14	1700	2.2712	2.25
6	5	15	2000	2.672	2.65
7	6	16	2300	3.0728	3.05
8	7	17	2600	3.4736	3.45
9	8	18	2900	3.8744	3.85
10	9	19	3200	4.2752	4.25
	10	20	3500	4.676	4.7
		21	3800	5.0768	5.1
		22	4100	5.4776	5.5
		23	4400	5.8784	5.9
		24	4700	6.2792	6.3
		25	5000	6.68	6.7
		26	5300	7.0808	7
		27	5600	7.4816	7.5
		28	5900	7.8824	7.9
		29	6200	8.2832	8.3

^1^ TN: the number of the treatment based on the milliliter method; ^2^ WA: the water quantity based on a single milliliter; ^3^ TN: treatment number based on the TKW method; ^4^ PW: the suggested percentage for the water quantity application in ml based on the TKW method; ^5^ WA: the water quantity equivalent to the suggested percentage of water volume based on TKW; ^6^ RAW: rounded quantity of water in ml to the nearest measurable digit on the pipette.

## Data Availability

All data, tables, and figures in this manuscript are original.

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
