# Peer review of "The Effects of Temperature and Water on the Seed Germination and Seedling Development of Rapeseed (Brassica napus L.)"

_plants, 2022, doi:10.3390/plants11212819_

Round 1

Reviewer 1 Report

This study investigated the effects of water and temperatures on seed germination and seedling growth on rapeseed. The results have revealed the optimal temperature and water range for rapeseed germination. And it is meaningful for the cultivation of the essential oilseed crop.

Some minor comments:

Line 122-124, delete

Line130, Table 3? The serial number of tables and pictures in the whole manuscript were orderless, check and revise it.

Line 395, “in vitro” italic

The quality of the pictures should be improved.

Author Response

Dear Reviewer,

Enclosed please find our answers for your review.

Sincerely,

The Authors

Reviewer 2 Report

In the study entitled “The Effect of Temperature and water on Seed Germination and Seedling Development of Rapeseed (Brassica napus L.)” hydroponic experiments were conducted to check the effects of water and temperatures on seed germination and seedling growth investigated on rapeseed crop (Brassica napus L. Allison).

The study topic is interesting and informative. However, the main concerns about this manuscript can be found below.

The authors should improve the overall English of the manuscript. The language is not satisfactory to be published at this stage in this journal.

The scientific names of the species and the names of the genes must be italicized in the manuscript. 

The abbreviations should be fully explained during the first mention in the abstract and introduction.

Many formatting mistakes have been found. I suggest authors review the whole manuscript carefully and correct all the mistakes.

Authors should update the references. Please cite recent references in the whole manuscript. Many old references have been found.

Please write Figures and tables in the brackets in the text.

Line 4: Add space between the names “Asma Haj Sghaier 1, Ákos Tarnawa 1, Hussein Khaeim 1 ,Gergő Péter Kovács 1 ,Csaba Gyuricza, and Zoltán Kende 1,*”

Line 18: 0,65 ml?

Lines 88-92: Please cite references

Line 100: Please italicize it “in vitro”

Lines 99-100: Please cite reference

Line 133: Please add space “10°C”. Keep consistency in the whole manuscript.

Line 161: Font size of “Figure 3 is different

Line 167: Change it to “(Figures 4-5 and Table 1- 2)

Line 201 and Line 247: Authors have labeled both figures as Figure 5. Please correct it.

Lines 247, 257, and 259: Please revise the figures. The bars on the graphs do not look natural.

Line 353: It should be “Figures 4 and 5”

Lines 409-457: Please italicize all the subheadings of the Materials and Methods section.

Line 494: Please use the same font style.

Some mistakes have been found in the references. Scientific names of the species and the names of the genes must be italicized. Please follow the instructions to the authors and set all the references according to the Journal’s format.

TRANSLATE with x English
Arabic Hebrew Polish
Bulgarian Hindi Portuguese
Catalan Hmong Daw Romanian
Chinese Simplified Hungarian Russian
Chinese Traditional Indonesian Slovak
Czech Italian Slovenian
Danish Japanese Spanish
Dutch Klingon Swedish
English Korean Thai
Estonian Latvian Turkish
Finnish Lithuanian Ukrainian
French Malay Urdu
German Maltese Vietnamese
Greek Norwegian Welsh
Haitian Creole Persian  
TRANSLATE with COPY THE URL BELOW Back EMBED THE SNIPPET BELOW IN YOUR SITE Enable collaborative features and customize widget: Bing Webmaster Portal Back

Author Response

(The authors gave the same response as above.)

Reviewer 3 Report

In the current manuscript, the authors examine the effects of water and temperature on the germination and growth of rapeseed plants. Germination was examined at different temperatures and water levels. Also planting densities and antifungal application techniques were analyzed in the study. Further, the authors intend their method of evaluation to be used in future studies on rapeseed germination, breeding programs, and predicting sowing dates.

1. The figure contains too many data lines, making it difficult for the reader to understand. Make a simple bar graph.

2. Include seed germination images if the authors have them.

3. all the figures need attention and redo the formatting.

4. Figure 5 looks incomplete. The scale of the Y-axis is missing.

5. Same comment, figure 6 and 7 looks incomplete

Author Response

(The authors gave the same response as above.)

Round 2

Reviewer 2 Report

After checking the responses from the authors and the revised version, the MS is ready for acceptance and publication.

TRANSLATE with x English
Arabic Hebrew Polish
Bulgarian Hindi Portuguese
Catalan Hmong Daw Romanian
Chinese Simplified Hungarian Russian
Chinese Traditional Indonesian Slovak
Czech Italian Slovenian
Danish Japanese Spanish
Dutch Klingon Swedish
English Korean Thai
Estonian Latvian Turkish
Finnish Lithuanian Ukrainian
French Malay Urdu
German Maltese Vietnamese
Greek Norwegian Welsh
Haitian Creole Persian  
TRANSLATE with COPY THE URL BELOW Back EMBED THE SNIPPET BELOW IN YOUR SITE Enable collaborative features and customize widget: Bing Webmaster Portal Back

Author Response

Dear Professor,

Thank you for your kind help in the evaluation process!

Sincerely yours,

The Authors

Reviewer 3 Report

The present manuscript can be accepted now.

Author Response

(The authors gave the same response as above.)
